# Visual Feedback for Self-Improving Text Layout with MLLM via Reinforcement Learning

## Abstract

Recent advances in Multimodal Large Language Models (MLLMs) have enabled automated generation of structured layouts from natural language descriptions. Existing methods typically follow a text-only paradigm that generates code to represent layouts, which are then rendered by graphic engines to produce final images. However, during the code generation process, they are blind to the rendered visual outcome, making it difficult to guarantee readability and aesthetics. In this paper, we identify visual feedback as a critical factor in layout generation and propose a self-improving framework that leverages visual feedback for text layout generation. Our method enables the model to iteratively generate layout code, render it into an image, visually evaluate the result, and refine the design through reflection until satisfactory quality is reached. We achieve this through reinforcement learning with a visually grounded reward model that incorporates OCR accuracy and aesthetic measures. Importantly, we demonstrate that simple outcome-based rewards are more effective than complex process-oriented reward functions for iterative generation tasks. Experiments across multiple benchmarks show that our approach significantly outperforms code-only baselines, advanced MLLMs, and existing layout models, establishing Visual Feedback as critical for design-oriented MLLMs.

## 1 Introduction

The emergence of Large Language Models(LLMs)(Achiam et al., 2023; DeepSeek-AI, 2025; Yang et al., 2025) and Multimodal Large Language Models (MLLMs)(Hurst et al., 2024; OpenAI, 2025; Bai et al., 2025; Team, 2025) has opened new possibilities for automated content generation tasks, particularly for structured visual layouts. These models can translate natural language descriptions directly into complex designs—such as typographic posters, social media graphics, and documents—by generating structured representations (e.g., SVG code or custom JSON)(Feng et al., 2024; Yang et al., 2024; Cheng et al., 2024; Qu et al., 2025) that specify the position, size, and style of each element (Jia et al., 2023; Inoue et al., 2024). Critically, MLLMs enhance this capability by leveraging cross-modal understanding, enabling them to condition the layout generation on not only textual prompts but also visual inputs.

However, existing methods face a fundamental limitation: they operate under a text-only paradigm that generate code to represent layout without visual feedback. For Instance, Jia et al. (2023) and Inoue et al. (2024) leverage LLMs to generate typography JSON files, while Zhang et al. (2025b) produces customized layout output formats, which are then composed into the final images by a graphic renderer. While these models can generate layout structures that conform to specifications, they lack the ability to directly perceive the visual appearance of their outputs. This limitation is critical because effective text layout design depends on intrinsically visual criteria such as aesthetic quality, text readability, and image-text coherence that cannot be fully captured by programmatic rules alone. For instance, a model may generate syntactically correct SVG code that results in overlapping elements, insufficient text-background contrast, or poor visual alignment, yet remain unaware of these visual defects.

Recent advances in Large Language Model(DeepSeek-AI, 2025; Jaech et al., 2024; OpenAI, 2025; Gandhi et al., 2025) research demonstrate that reflection, backtracking, and self-validation mechanisms can substantially improve performance on complex reasoning tasks. Moreover, Reinforce-

ment Learning(RL)(Ouyang et al., 2022a; Schulman et al., 2017; Shao et al., 2024) techniques have proven effective in activating the reflective reasoning capabilities of LLMs. This motivates our core research question: Can such reasoning capabilities be transferred to text layout generation to overcome the visual perception gap in existing approaches? We argue that the solution lies in incorporating **Visual Feedback** into the text layout generation process, leveraging MLLMs' inherent cross-modal understanding capabilities. Our key insight is that models should not only generate layout code but also perceive the rendered results to evaluate quality, diagnose visual issues, and devise optimization strategies through iterative refinement.

In this paper, we propose a novel **Self-Improving framework** for text layout generation that establishes a closed-loop process guided by reinforcement learning. As shown in Figure 1, the framework works as follows: the MLLM first generates initial SVG layout code, which is rendered into a visual image. This rendered image is fed back to the same model for visual inspection and reflection. If issues are identified, the model generates revised code and repeats the process, creating a continuous loop of "*generation, rendering, reflection and refinement*" until a satisfactory layout is achieved. Our approach employs a two-stage training framework. First, we construct a dataset of multi-stage generation–reflection–refinement trajectories using an advanced MLLM, followed by Supervised Fine-Tuning (SFT) to initialize the model for iterative generation. Second, we employ reinforcement learning to enhance the model's reflective capabilities, using a reward model that evaluates layout quality holistically and incorporates text accuracy through Optical Character Recognition (OCR). Our results demonstrate that MLLMs' visual understanding capabilities can be effectively activated through simple reward signals, enabling robust iterative improvement through visual feedback.

We conduct extensive experiments on the Qwen2.5-VL-7B(Bai et al., 2025) model for the task of layout target text on background images. Both quantitative and qualitative evaluations show that our visual feedback-driven method significantly outperforms code-only baselines and state-of-the-art layout generation approaches, while also surpassing advanced MLLMs and image editing models. This work establishes visual feedback as a critical component in generative text layout and provides a practical framework for developing self-improving MLLM-based design agents. Beyond achieving state-of-the-art performance, our experiments also reveal an important insight into reward design: **simple outcome-based rewards are not only sufficient to activate visual self-improvement, but also outperform more complex process-oriented supervision.** This finding highlights that stable and effective reflection can emerge without complex process signals, simplifying the application of reinforcement learning to design-oriented tasks.

The contributions of this work can be summarized as follows:

- **Problem identification**: We identify the critical limitation of existing code-based layout methods—the absence of visual feedback prevents effective output evaluation and optimization of vision-related performance.
- **Novel framework**: We propose a **self-improving framework** that equips MLLMs with a generation, rendering, reflection, refinement cycle, enabling iterative layout refinement guided by **visual feedback**. To the best of our knowledge, this is the first work that introduces such a visual feedback loop for layout generation.
- **Training methodology**: We design a two-stage SFT+RL pipeline and demonstrate that iterative reflective generation can be activated using simple reward signals from final outputs, without complex intermediate reward engineering.
- **Empirical validation**: We establish the effectiveness of our approach through comprehensive experiments, providing a practical framework for MLLM-based graphic design applications.

## 2 RELATED WORK

### 2.1 MULTIMODAL LARGE LANGUAGE MODEL

Recent progress in Multimodal Large Language Models (MLLMs) is driven by integrating pre-trained vision encoders (Radford et al., 2021; Zhai et al., 2023) with LLMs. The two modalities are typically aligned via lightweight projectors or Q-Former (Li et al., 2023) structures, a paradigm that has spurred a suite of powerful models. This includes open-source series like LLaVA (Liu et al.,

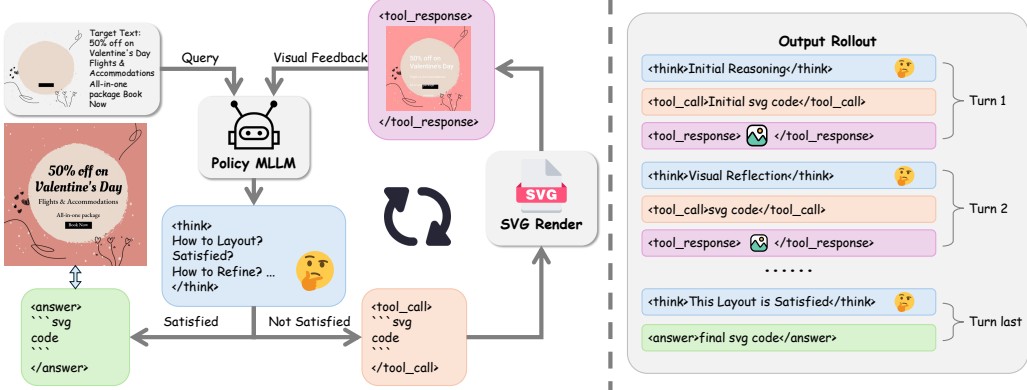

Figure 1: Our Visual Feedback framework: the left subgraph shows the iterative generation, rendering, reflection, and refinement cycle of our model; the right subgraph displays the multi-round data output by our model rollout.

2023; 2024), Qwen-VL (Bai et al., 2023; 2025), and Intern-VL (Chen et al., 2024b; Wang et al., 2025), as well as large-scale proprietary systems such as GPT-4o (Hurst et al., 2024), Gemini (Team et al., 2023), and Claude (Anthropic, 2025), which continue to advance the state of the art through massive scaling and enhanced reasoning techniques (Wei et al., 2022; Zhang et al., 2025c).

## 2.2 GRAPHIC LAYOUT GENERATION

Graphic layout generation has rapidly evolved from early generative models (GANs, VAEs) and Transformer-based architectures (Zhou et al., 2022; Lin et al., 2023b) to methods centered on LLMs. Current approaches leverage the reasoning and code-generation capabilities of these models. While some works utilize multimodal cues (Yang et al., 2024) or hierarchical generation structures (Cheng et al., 2024), a dominant trend is to frame layout creation as a code generation task. These methods prompt LLMs to output structured, language-based representations such as SVG, JSON, or other custom formats (Lin et al., 2023a; Seol et al., 2024; Chen et al., 2024a; Jia et al., 2023).

## 2.3 REINFORCEMENT LEARNING

Reinforcement learning is a cornerstone for aligning LLMs with human preferences, standardized by the RLHF (Ouyang et al., 2022a) pipeline which typically uses Proximal Policy Optimization (PPO) (Schulman et al., 2017). To mitigate the instability and high cost associated with PPO, recent alternatives like Direct Preference Optimization (DPO) (Rafailov et al., 2023) and Group Relative Policy Optimization (GRPO) (Shao et al., 2024) offer more direct and efficient optimization strategies. This alignment paradigm extends naturally to multimodal settings to improve visual grounding and reasoning. Works such as Vision-R1 (Huang et al., 2025), R1VL (Zhang et al., 2025a), and DeepEyes (Ziwei Zheng, 2025) adapt RL to vision-language models by incorporating multimodal rewards, chain-of-thought signals, and specialized replay mechanisms, demonstrating the power of RL in enhancing multimodal alignment and capability.

## 3 METHOD

### 3.1 TASK FORMULATION

This work aims to develop a self-improving agent for text layout generation that optimizes outputs through visual feedback. Our approach employs a two-stage training framework: (1) Cold-Start SFT to equip the model with basic iterative generation and reflection capabilities, and (2) Reinforcement Learning to enhance performance using vision-based reward signals.

As shown in Figure 1, we formulate the task as a multi-round interaction between the model and a rendering environment. Given a background image and target text, the model follows an iterative cycle of *generation, rendering, reflection, refinement*:

1. **Initial Generation:** The model first analyzes the input through reasoning, then generates initial layout code through a structured tool call.

2. **Rendering:** The rendering tool converts the SVG code into a visual layout image and feeds it back to the same MLLM.

3. **Visual Reflection:** The model examines the rendered layout visual image through reasoning to evaluate whether the quality is satisfactory to it.

4. **Iterative Refinement:** If unsatisfied, the model reasons about necessary modifications and generates a revised layout code, repeating until the model determines satisfaction with the layout quality.

This process is formalized in Algorithm 1.

## 3.2 STAGE 1: COLD-START SUPERVISED FINE-TUNING (SFT)

The cold-start SFT stage enables the model to acquire iterative behavior, self-reflection capabilities, and tool usage specifications through distillation from a powerful teacher model.

**Data Construction for Iterative Reflection** Due to the absence of natural multi-round reflection data, we employ Doubao-Seed-1.6(ByteDance, 2025) as a teacher model for data synthesis through a two-step process:

1. **Initial Generation Synthesis:** We prompt Doubao-Seed-1.6 with background images and ground-truth layouts to generate reasoning processes for SVG code generation. We then fine-tune Qwen2.5-VL-7B on these data and collect its inference outputs, which serve as suboptimal initial attempts for subsequent reflection synthesis.

2. **Multi-Round Reflection Synthesis:** We input the initial attempts from Step 1 along with ground-truth layouts to Doubao-Seed-1.6, instructing it to perform iterative reflection and modification to reach the ground-truth solution. This simulates realistic human design refinement processes and generates complete multi-round reflection trajectories.

We combine the synthesized data from Step1 and Step2 and organize them using structured tags: the intermediate rounds use `<think>` and `<tool_call>` tags, while the final round uses `<think>` and `<answer>` tags containing the completed layout. The right subfigure in Figure 1 shows specific data examples. For specific data distillation prompts, see Appendix G.

**Training Objective** We fine-tune Qwen2.5-VL-7B using causal language modeling on the synthesized dialogue data. To prevent the model from learning suboptimal outputs, we mask the loss for initial responses in improvement sequences, ensuring the model learns from the correction process rather than initial errors.

## 3.3 STAGE 2: REINFORCEMENT LEARNING (RL)

### 3.3.1 RL ALGORITHM

We adopt the Group Relative Policy Optimization (GRPO)(Shao et al., 2024) algorithm for reinforcement learning and make certain improvements to the advantage function. Compared with traditional policy optimization methods, GRPO performs policy gradient optimization within sample groups, enabling the model to learn in the direction of maximizing rewards. The optimization objectives of GRPO are as follows:

$$
\mathcal{J}_{\text{GRPO}}(\theta) = \mathbb{E}\big[q \sim P(Q), \{o_i\}_{i=1}^{G} \sim \pi_{\theta_{\text{old}}}(O \mid q)\big]
$$

$$
\frac{1}{G}\sum_{i=1}^{G}\frac{1}{|o_i|}\sum_{t=1}^{|o_i|}\Big\{\min\Big[\frac{\pi_\theta(o_{i,t}\mid q, o_{i,<t})}{\pi_{\theta_{\text{old}}}(o_{i,t}\mid q, o_{i,<t})}A_{i,t},
$$

$$
\text{clip}\Big(\frac{\pi_\theta(o_{i,t}\mid q, o_{i,<t})}{\pi_{\theta_{\text{old}}}(o_{i,t}\mid q, o_{i,<t})}, 1-\varepsilon, 1+\varepsilon\Big)A_{i,t}\Big] - \beta\,\mathbb{D}_{\text{KL}}\big[\pi_\theta \,\|\, \pi_{\text{ref}}\big]\Big\},
$$

(1)

where $\epsilon$ and $\beta$ are the clipping hyperparameters and the KL divergence penalty coefficient, respectively. Subsequently, we elaborate on our approaches to computing the reward function and the advantage function.

We design a three-component reward function to score the layout effect: (1) $R_{\text{layout}}$(Section 3.3.2), a specialized reward model trained to evaluate overall layout quality; (2) $R_{\text{ocr}}$, a text accuracy reward based on OCR recognition. Specifically, we run OCR on the rendered layout and compute the character-level accuracy between the recognized string and the target text; and (3) $R_{\text{svg}}$, a code-level accuracy reward calculated by comparing text strings extracted from the SVG file with the target text. The total reward for the layout effect is the weighted sum of the three components:

$$R_{\text{score}} = R_{\text{layout}} + \alpha \cdot (R_{\text{ocr}} + R_{\text{svg}}), \tag{2}$$

where $\alpha$ balance aesthetic quality against functional accuracy. In addition, we incorporate a format reward (Equation 3) to constrain the output format of the model:

$$R_{\text{format}} = \begin{cases} 1.0, & \text{if format is correct,} \\ -1.0, & \text{if format is incorrect.} \end{cases} \tag{3}$$

Due to the multi-round nature of our approach, format rewards are applied separately in each round, leading to inconsistent rewards across rounds. Following Hu et al. (2025), we use the mean value of $R_{\text{score}}$ within the group as a baseline to reshape the reward, and then add the format reward:

$$A = R_{\text{score}} - mean_{\text{group}}(R_{\text{score}}) + \gamma \cdot R_{\text{format}}, \tag{4}$$

where $\gamma$ is a hyperparameter that controls the balance between the layout effect reward and the format reward. Finally, normalize advantages across the global batch, which are used for training:

$$A^{norm} = \frac{A - \text{mean}_{batch}(A)}{\text{std}_{batch}(A)}. \tag{5}$$

### 3.3.2 REWARD MODEL TRAINING

To obtain $R_{\text{layout}}$, we train a specialized reward model that takes triplets $(B, T, I)$ as input, where $B$ denotes the background image, $T$ denotes the target text, and $I$ denotes the rendered layout image. The model then outputs a scalar score that assesses the overall layout quality.

Following the method proposed in Ouyang et al. (2022b), we initialize the reward model using Qwen2.5-VL-3B. To adapt the model for preference learning, we replace the final layer with a linear layer that produces a scalar output. Subsequently, the reward model is then trained using the negative log-likelihood loss function:

$$\mathcal{L}_{\text{RM}}(\theta) = -\mathbb{E}_{(q,o^+,o^-)\sim\mathcal{D}} \left[ \log \sigma \left( r_\theta \left( q, o^+ \right) - r_\theta \left( q, o^- \right) \right) \right]. \tag{6}$$

**Dataset:** A high-quality preference dataset is paramount for training a robust reward model that can guide reinforcement learning without succumbing to reward hacking. However, no existing datasets or methodologies are specifically designed for layout generation reward modeling. To address this gap, we introduce a novel hierarchical data construction methodology that creates fine-grained quality distinctions across multiple layout quality levels.

Our methodology constructs four distinct quality levels to capture fine-grained layout performance:

- **Level-I:** High-quality ground-truth layouts serving as gold standards for design excellence.

- **Level-II:** Layouts generated by Qwen2.5-VL-7B after 5 epochs of fine-tuning on 200K samples, exhibiting reasonable quality with minor imperfections.

- **Level-III:** Layouts produced by early training checkpoints with systematic spatial perturbations applied to layout elements, including random positional offsets that moderately compromise design coherence.

- **Level-IV:** Severely degraded layouts from the same early checkpoints subjected to aggressive perturbations: extensive positional displacement, random font size variations, selective text element deletion, image reference removal, and arbitrary SVG scaling transformations.

This hierarchical construction enables comprehensive preference learning through systematic pairwise comparisons. For each layout generation prompt, we create layouts at all four quality levels, then form all possible pairwise comparisons between different levels. This yields $\binom{4}{2} = 6$ preference pairs per problem, establishing clear quality orderings that capture nuanced distinctions essential for effective reward model training. This strategy forces the reward model to move beyond simple binary (good/bad) judgments and learn the subtle distinctions that separate excellent layouts from merely acceptable ones. The resulting dataset provides a comprehensive and reliable basis for training a highly discerning layout reward model, $r_\theta$. For the evaluation of the reward model, please refer to the Appendix E.3.

## 4 EXPERIMENT

### 4.1 EXPERIMENTAL SETUP

**Benchmark Test Sets** For evaluation, we randomly sample 1K examples from our dataset as the primary test set, ensuring no overlap with training data. We also conduct additional experiments on the Crello(Yamaguchi, 2021) and DESIGNERINTENTION(Jia et al., 2023) benchmarks, preprocessed into background-text pairs. The results for these additional benchmarks are shown in Appendix E.

**Evaluation metrics** We adopt three groups of evaluation metrics. Text accuracy is measured using character-level precision, recall, and F-measure based on OCR recognition. Layout quality is assessed with $R_{ali}$, $R_{ove}$, and $R_{com}$ (Zhou et al., 2022), which capture cross-modal alignment, text-text overlap, and pixel gradient smoothness within text regions. Additionally, we employ GPT-4o as a judge to evaluate four dimensions: Text Accuracy, Text-Background Harmony, Text Presentation Quality, and Meaning Expression Adaptability. For Details, please refer to Appendix.

**Baselines** We compare against three categories of baselines: (1) **Advanced MLLMs** including GPT-4o, Claude 3.7, Doubao-Seed-1.6, and Qwen2.5-VL-72B; (2) **Image editing models** covering GPT-4o(edit), Qwen-Image-Edit(Wu et al., 2025), and FLUX-Kontext(Labs et al., 2025); (3) **Specialized layout generation** using the open-source domain-specific model OpenCOLE(Inoue et al., 2024).

### 4.2 COMPARISON WITH EXISTING METHODS

Table 1: The Graphic quality metrics and OCR metrics on test set, where **Visual Feedback-step1** and **Visual Feedback-answer** respectively represent the metrics of our results in the first output and the final result output after iterative reflection.

| Method | Model | Param Size | OCR | | | Graphic | | | RM Score ↑ |
| --- | --- | --- | --- | --- | --- | --- | --- | --- | --- |
| | | | Char-P ↑ | Char-R ↑ | Char-F ↑ | $R_{ali} \downarrow$ | $R_{ove} \downarrow$ | $R_{com} \downarrow$ | |
| MLLM | GPT-4o | - | 0.9076 | 0.7575 | 0.8258 | 0.0046 | 0.0033 | 18.8443 | 0.3561 |
| | Claude3.7 | - | 0.9295 | 0.8127 | 0.8672 | 0.0053 | 0.0383 | 16.4401 | 0.5295 |
| | Doubao-Seed-1.6 | 230B | 0.9215 | 0.7860 | 0.8484 | 0.0058 | 0.0216 | 18.5823 | 0.4063 |
| | Qwen2.5-VL | 72B | 0.7910 | 0.6571 | 0.7178 | **0.0031** | 0.0358 | 19.9399 | 0.1989 |
| | Qwen2.5-VL | 7B | 0.7618 | 0.5220 | 0.6195 | **0.0029** | 0.0229 | 25.7715 | 0.1166 |
| Image Edit | GPT-4o | - | 0.7731 | 0.6734 | 0.7198 | - | - | - | 0.3371 |
| | Qwen-Image-Edit | 20B | 0.7165 | 0.7349 | 0.7256 | - | - | - | 0.3062 |
| | FLUX Kontext | 12B | 0.2207 | 0.0944 | 0.1322 | - | - | - | -0.5853 |
| Layout | OpenCOLE | 7B | 0.4041 | 0.1462 | 0.2147 | 1.2029 | 0.0316 | 25.1150 | 0.0397 |
| Ours | Visual Feedback-step1 | 7B | 0.9619 | 0.8593 | 0.9071 | 0.0035 | 0.0059 | 15.4583 | 0.5415 |
| | Visual Feedback-answer | 7B | **0.9675** | **0.9096** | **0.9376** | 0.0039 | **0.0009** | **11.8678** | **0.6018** |
| | Δ (vs step1) | 7B | +0.0056 | +0.0503 | +0.0305 | -0.0004 | +0.005 | +3.5905 | +0.0603 |

As shown in Table 1, Tabel 2 and Figure 2, our Visual Feedback method consistently outperforms all baselines across three distinct categories of metrics by a significant margin. On OCR metrics, the F1 score of our model's initial generation (0.9071) is already substantially higher than the second-best performer, Claude3.7 (0.8672). After optimization via visual feedback, this performance gap is further widened with an improvement of +0.0305. In terms of Graphic metrics, our method achieves competitive performance on the alignment metric ($R_{ali}$) and demonstrates a remarkable advantage in minimizing element overlap ($R_{ove}$) and optimizing text composition ($R_{com}$). The benefit of the visual feedback is particularly pronounced in other areas, contributing to the overall superior performance. This trend is mirrored by the reward model scores, where both our initial and final outputs achieve the highest scores among all methods. On the comprehensive GPT-4o evaluation, our visual feedback mechanism yields improvements across all four dimensions, with both the initial and

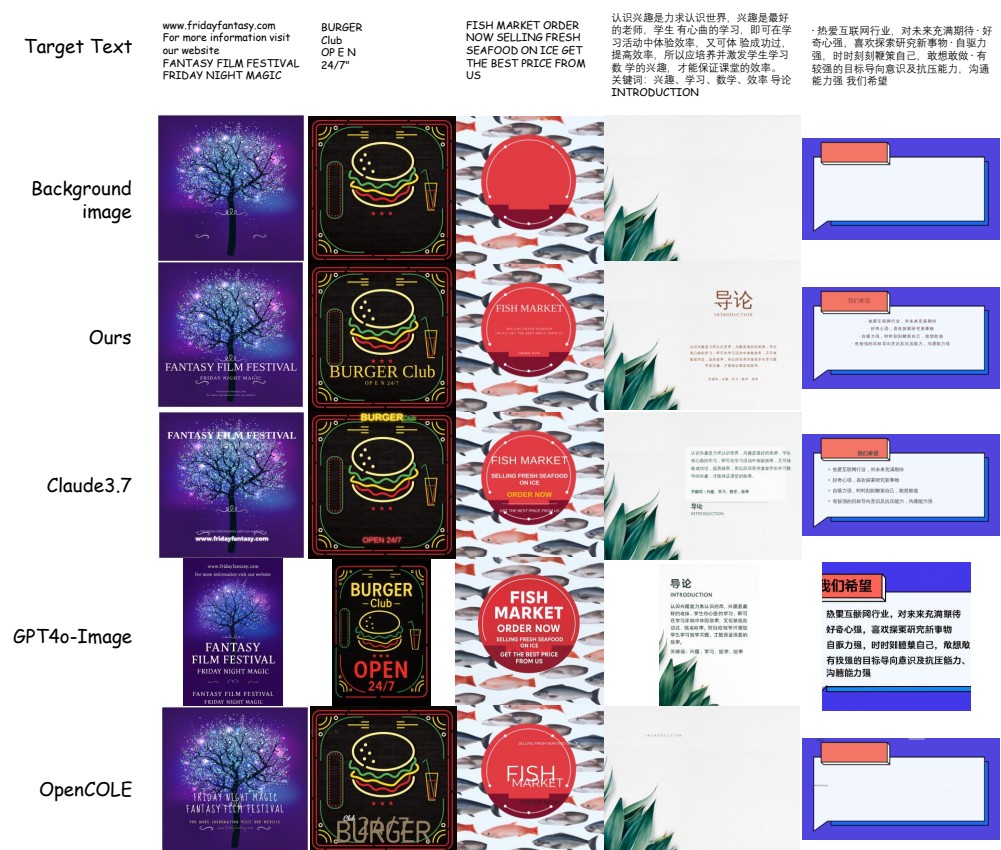

Figure 2: In comparison with existing methods, we selected one model from each category of methods as a representative. For a more comprehensive comparison, please refer to the Figure 4.

Table 2: The GPT-4o metrics on test set. The value range of each evaluation dimension is between 0 and 10, in which **Overall** represents the average of the scores across four dimensions.

| Method | Model | Param Size | GPT-4o | | | | |
|---|---|---|---|---|---|---|---|
| | | | Text | Harmony | Quality | Meaning | Overall |
| | GPT-4o | - | 8.5165 | 8.0341 | 7.2826 | 7.6553 | 7.8721 |
| | Claude3.7 | - | 8.8058 | 8.4026 | 7.7691 | 8.2651 | 8.3106 |
| MLLM | Doubao-Seed-1.6 | 230B | 8.5663 | 8.2304 | 7.4463 | 7.8917 | 8.0337 |
| | Qwen2.5-VL | 72B | 7.8633 | 7.6094 | 6.5402 | 6.5582 | 7.1428 |
| | Qwen2.5-VL | 7B | 7.5638 | 6.4431 | 5.4827 | 5.3861 | 6.5489 |
| | GPT-4o | - | 7.4809 | **8.9298** | **8.1672** | 8.1919 | 8.1924 |
| Image Edit | Qwen-Image-Edit | 20B | 5.4086 | 7.7783 | 6.1886 | 6.0040 | 6.3449 |
| | FLUX Kontext | 12B | 1.6473 | 6.8098 | 3.6345 | 2.2142 | 3.5765 |
| Layout | OpenCOLE | 7B | 2.6596 | 6.3873 | 3.7020 | 2.8896 | 3.9096 |
| | Visual Feedback-step1 | 7B | 8.8880 | 8.3591 | 7.7255 | 8.2896 | 8.3155 |
| Ours | Visual Feedback-answer | 7B | **9.0447** | _8.7492_ | _7.9679_ | **8.5969** | **8.5897** |
| | Δ (vs step1) | 7B | +0.1567 | +0.3901 | +0.2424 | +0.3073 | +0.2742 |

final results establishing state-of-the-art performance on overall metrics. In the qualitative comparison shown in Figure 2, our model demonstrates performance that is highly competitive with the state-of-the-art Claude3.7 model. The advantage over the open-source layout model, OpenCOLE, is obvious. Furthermore, representative image editing methods struggle with dense text, especially in Chinese, and often inevitably alter the background image. This introduces undesirable artifacts and fundamentally conflicts with our primary task of text layout.

### 4.3 ABLATION STUDY

In the ablation study, we conducted comparative experiments using our constructed test set. To demonstrate the advantages of our iterative Visual Feedback method, we compared it against several training approaches: (1) **Cold-Start Model**: The baseline model mainly ensures the format of iterative outputs and the syntactic validity of SVG code, but it does not significantly improve the layout quality; (2) **Single-Round RL**: We trained the cold-start model using RL but restricted generation to only one step, enabling fair comparison between single-round generation and iterative reflection; (3) **RL from Pre-trained Models**: Direct RL training from the pre-trained Qwen2.5-VL-7B model without SFT initialization; (4) **Direct Output SFT+RL**: SFT+RL training for direct SVG code generation using the same source data as our Visual Feedback method; (5) **Direct SFT**: For fair comparison with our 40K-sample SFT+RL approach, we trained a direct SFT model on 40K samples. Please refer to the Appendix D for the training details of all models.

Table 3: Multiple ablation experiments on Graphic quality metrics and OCR metrics on our test set.

| Model | OCR | | | Graphic | | | RM Score ↑ |
|---|---|---|---|---|---|---|---|
| | Char-P ↑ | Char-R ↑ | Char-F ↑ | $R_{ali}$ ↓ | $R_{ove}$ ↓ | $R_{com}$ ↓ | |
| Visual Feedback-step1 | 0.9619 | 0.8593 | 0.9071 | 0.0035 | 0.0059 | 15.4583 | 0.5415 |
| Visual Feedback-answer | **0.9675** | **0.9096** | **0.9376** | 0.0039 | **0.0009** | **11.8678** | **0.6018** |
| cold-start-step1 | 0.9287 | 0.6894 | 0.7913 | 0.0078 | 0.0297 | 19.0560 | 0.2572 |
| cold-start-answer | 0.9306 | 0.6985 | 0.7980 | 0.0081 | 0.0298 | 19.0577 | 0.2608 |
| cold-start-rl-only-step1 | 0.9422 | 0.8242 | 0.8792 | 0.0024 | 0.0053 | 18.9428 | 0.4063 |
| Qwen2.5-VL-7B | 0.7618 | 0.5220 | 0.6195 | 0.0029 | 0.0229 | 25.7715 | 0.1166 |
| direct RL | 0.8865 | 0.7549 | 0.8154 | **0.0003** | 6.7109 | 22.0273 | 0.2971 |
| direct sft8k+rl | 0.9606 | 0.8895 | 0.9237 | 0.0027 | 0.0021 | 17.0654 | 0.4964 |
| basemodel-direct-40k | 0.9150 | 0.8025 | 0.8551 | 0.0040 | 0.0153 | 12.9459 | 0.5332 |

The results of our ablation study, presented in Table 3, demonstrate the clear superiority of our Visual Feedback method. Our model (Visual Feedback-answer) achieves the best performance across the majority of metrics, substantially outperforming all reinforcement learning baselines. More notably, the quality of our initial output (Visual-Feedback-step1) is already not inferior to any other competing methods. For instance, its RM Score of 0.5415 surpasses even the strong basemodel-direct 40k (0.5332). Subsequent iterative steps further widened this performance gap, increasing the RM score to 0.6018, and achieving top-notch results in both OCR and image quality. Our success highlights that our visual feedback framework is a more effective solution for layout generation, as it can establish a higher quality benchmark from the very first step and then optimize it to the state-of-the-art level.

## 5 DISCUSSION

**Can simple outcome-based rewards effectively stimulate self-improvement capabilities in MLLMs?** Our empirical investigation provides compelling evidence that they can, and even outperform more sophisticated alternatives.

Recent research in agentic RL(Singh et al., 2025; Dong et al., 2025) typically employs complex, fine-grained reward functions to guide specific capabilities. To rigorously evaluate this paradigm, we designed and implemented a sophisticated process-oriented reward function in our Visual Feedback framework (Equation 7–9). This complex reward scheme incorporates three distinct optimization objectives: (1) first-round quality maximization using group-wise mean baselines, (2) iterative improvement encouragement through maximum-quality baselines from previous rounds, and (3) strategic termination control via reward bonuses and length penalties to prevent premature convergence and reward hacking behaviors. Subsequent advantages will be normalized through Equation 5.

$$A_{q,o_{t_i}} = \begin{cases} R_{q,o_{t_1}} - \text{mean}_{\text{group}}(R_{q,o_{t_1}}) & \text{if } i = 1, \\ 2 \cdot \left( R_{q,o_{t_i}} - \max(R_{q,o_{t_{\leq i}}}) \right) & \text{if } i \neq 1 \text{ and } i \neq \text{last}, \\ 2 \cdot (R_{\text{answer}} + R_{\text{length}}) & \text{if } i = \text{last}. \end{cases} \quad (7)$$

where $o_t$ represents a complete generation path, and $o_{t_i}$ represents the response of the $i$-th round in this complete path; $i = \text{last}$, the terminal reward components are defined as:

$$R_{\text{answer}} = \begin{cases} 0.7 & \text{if } R_{q,o_{t_{\text{last}}}} \geq \max(R_{q,o_{t_{\leq \text{last}}}}), \\ R_{q,o_{t_{\text{last}}}} - \max(R_{q,o_{t_{\leq \text{last}}}}) & \text{else,} \end{cases} \quad (8)$$

$$R_{\text{length}} = -2 \cdot \left( \left( R_{q,o_{t_{\text{last}}}} - \max_{\text{group}}(R_{q,o_{t_{\text{last}}}}) \right) \cdot \max(0, 4 - \text{tool\_call\_count}) \right), \qquad (9)$$

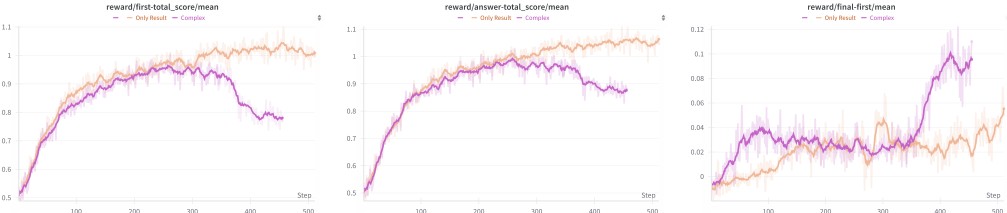

Figure 3: Comparison of training processes: simple outcom rewards vs. complex process rewards.

Figure 3 illustrates the training dynamics of the two reward schemes. Before 250 steps, both algorithms improved stably: their final answer scores (middle subfigure) showed nearly identical trends, and first-round generation performance was comparable, the outcome-only reward algorithm even slightly outperform the Agentic RL one. After 250 steps, the Agentic RL algorithm converged and even suffered performance degradation, whereas the outcome-only reward algorithm continued to improve steadily until training concluded. The right subfigure (difference between final output score and first-round score) reveals that Agentic RL quickly widened this gap but plateaued later—likely due to restricted first-round learning in early training. In contrast, the outcome-only reward algorithm gradually mastered progressive iterative refinement.

Table 4: Comparison of simple outcome rewards and complex process rewards on our test set.

| level | OCR | | | Graphic | | | RM Score ↑ |
|---|---|---|---|---|---|---|---|
| | Char-P ↑ | Char-R ↑ | Char-F ↑ | $R_{ali}$ ↓ | $R_{ove}$ ↓ | $R_{com}$ ↓ | |
| Only Outcome RL-step1 | 0.9619 | 0.8593 | 0.9071 | **0.0035** | 0.0059 | 15.4583 | 0.5415 |
| Only Outcome RL-answer | **0.9675** | **0.9096** | **0.9376** | 0.0039 | **0.0009** | **11.8678** | **0.6018** |
| Δ (vs step1) | +0.0056 | +0.0503 | +0.0305 | -0.0004 | +0.005 | +3.5905 | +0.0603 |
| Agentic RL-step1 | 0.9538 | 0.8577 | 0.9032 | 0.0053 | 0.0030 | 16.6528 | 0.4936 |
| Agentic-RL-answer | **0.9693** | **0.8825** | **0.9239** | 0.0052 | 0.0027 | 16.4220 | 0.5241 |
| Δ (vs step1) | +0.0155 | +0.0248 | +0.0207 | +0.0001 | +0.0003 | +0.2308 | +0.0305 |

Table 4 quantifies these observations through comprehensive evaluation metrics. Despite marginal differences in first-round generation quality, our simple outcome-based reward demonstrates superior effectiveness in stimulating self-improvement capabilities across all evaluation dimensions.

These findings reveal a fundamental insight: under effective visual feedback mechanisms, simple outcome-based rewards can successfully harness the inherent visual understanding capabilities of multimodal models to elicit robust self-improvement behaviors, while complex process-oriented rewards may actually inhibit optimal performance. This counterintuitive result suggests that the powerful internal representations and reasoning capabilities of modern MLLMs, when properly guided by clear outcome objectives and visual feedback, can autonomously develop sophisticated iterative refinement strategies without explicit process supervision.

## 6 CONCLUSION

We have introduced a self-improving framework that successfully bridges the gap between code generation and visual perception in text layout design. Our method empowers MLLMs to progressively enhance their own creations through a visual feedback loop of iterative generation, rendering, reflection, and refinement. A key finding is that this self-improvement can be driven by simple, outcome-based rewards, circumventing the need for complex reward engineering. Extensive experiments validate our approach, demonstrating that it not only surpasses specialized layout models but also outperforms state-of-the-art MLLMs and image editing systems. Ultimately, this work establishes visual feedback as a critical component for high-quality automated design and charts a clear course toward more autonomous, self-improving creative agents.

ETHICS STATEMENT

Our research adheres to the ICLR Code of Ethics. We acknowledge that our models, trained on public datasets, may inherit societal biases, and we recognize the dual-use nature of this technology for potential misuse. In the spirit of transparency and reproducibility, we plan to publicly release our datasets, models, and reward model. To mitigate risks and encourage responsible innovation, this release will be accompanied by a detailed model card outlining the system's capabilities and limitations, as well as a license restricting use in malicious, deceptive, or exploitative applications.

REPRODUCIBILITY STATEMENT

We are committed to ensuring the full reproducibility of our research. To this end, we will release our complete code for data processing and all training stages in the future. This includes the implementation of our hierarchical data construction process for the reward model, as detailed in Section 3.3.2. Key training hyperparameters for supervised fine-tuning, reward model training, and reinforcement learning are described in detail in Appendix E.3 and D. Furthermore, the weights for our final model and the reward model will be made publicly available to facilitate verification and future research. We believe these resources provide a comprehensive basis for the community to reproduce our findings and build upon our work.

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

APPENDIX

# A USAGE OF LARGE LANGUAGE MODELS

The authors acknowledge the use of Large Language Models (LLMs) in the preparation of this manuscript to enhance its clarity. The LLM served solely as a general-purpose writing assistant, with its role strictly limited to improving the language and presentation of our manuscript. Specifically, the LLM was used to rephrase sentences for greater clarity and fluency, as well as to correct grammatical and spelling errors.

It is important to state that all scientific contributions—including the core research ideas, methodology, technical implementation, experimental results, and scientific findings—are the original work of the authors. The LLM had no role in the conception or execution of the research. We take full responsibility for the final content of this paper.

# B VISUAL FEEDBACK ALGORITHM

---

**Algorithm 1** Visual Feedback for Layout Self-Improvement

---

**Require:** Background image $I_b$, target text $T$
**Ensure:** Final layout code $S_{\text{final}}$
1: $S \leftarrow$ Model generates initial layout via tool call based on $I_b$ and $T$
2: $I_{\text{rendered}} \leftarrow \text{Render}(S)$
3: **while** Model determines dissatisfaction **do**
4:     Reflection $\leftarrow \text{Model}(\text{reason}|I_{\text{rendered}})$
5:     **if** Reflection indicates satisfaction **then**
6:         **return** $S$ as the final result
7:     **else**
8:         $S \leftarrow \text{Model}(\text{tool call}|\text{Reflection})$         ▷ Generate revised layout
9:         $I_{\text{rendered}} \leftarrow \text{Render}(S)$
10:     **end if**
11: **end while**

---

# C DATASETS AND EVALUATION METRICS

We use an OCR engine[1] to recognize text in design images and evaluate the accuracy of rendered text using character-level precision, recall, and f-measure. In the reinforcement learning reward function, $R_{\text{ocr}}$ and $R_{\text{svg}}$ are evaluated using accuracy. Specifically, a character in the OCR recognition result is defined as a True Positive (TP) if it appears in the annotation; otherwise, it is classified as a False Positive (FP). A False Negative (FN) indicates that a character is only present in the annotation but absent from the OCR recognition result. Accordingly, character-level precision, recall, f-measure and accuracy can be formulated as follows:

$$Char\_P = \frac{TP}{TP + FP}, \qquad\qquad Char\_R = \frac{TP}{TP + FN},$$
$$Char\_F = \frac{2 \times Char\_P \times Char\_R}{Char\_P + Char\_R}, \qquad Char\_Acc = \frac{TP}{TP + FP + FN}. \tag{10}$$

The detailed prompts for evaluation using GPT-4o are shown blew. It conducts a comprehensive evaluation from four dimensions: **Text Accuracy** evaluates if the text content is perfectly correct and free of any errors. **Text-Background Harmony** assesses how well the text is visually integrated with the background for clarity and composition. **Text Presentation Quality** judges the intrinsic readability and structural organization of the text elements themselves. **Meaning Expression Adaptability** measures how effectively the text's design reinforces its message, tone, and contextual meaning.

---

[1] https://github.com/PaddlePaddle/PaddleOCR

```
GPT-4o Evaluation Prompt

You are an autonomous AI Assistant specializing in evaluating the
↪  typesetting effects of a typesetting model. This model's core
↪  task is to typeset user-specified text on a background image;
↪  your goal is to provide objective, targeted, and constructive
↪  scoring and feedback based on text-typesetting-specific
↪  principles and practical application needs. Your evaluation
↪  covers four independent dimensions: text content accuracy,
↪  text-background visual harmony, text presentation quality, and
↪  meaning expression adaptability. You will be provided with the
↪  background image, the user's original specified text, and the
↪  typeset result (background image + typeset text). Your task is
↪  to score the typesetting effect objectively based on the
↪  following 4 criteria and provide concise reasoning for each
↪  score.

Scoring rules:
- For each of the 4 criteria, score objectively and rigorously on an
↪  independent scale of 1-10. For a single criterion, a score of 10
↪  means flawless performance (no issues, fully meeting
↪  expectations); a score of 7 indicates minor flaws (no impact on
↪  core performance); a score of 4 reflects significant
↪  shortcomings (affecting core performance); a score of 1-2
↪  signifies severe issues (rendering the function of this
↪  criterion ineffective).
- Keep reasoning concise (1-2 sentences per criterion), focusing on
↪  specific performance. If the output is too long, it will be
↪  truncated.
- Only respond in JSON format with 4 top-level keys corresponding to
↪  the 4 Grading criteria. Each key's value is an object containing
↪  "score" (integer 1-10) and "reason" (string). No other
↪  information.

Grading criteria:

1. Text Accuracy (1-10): Evaluate consistency with the user's
↪  original text (no missing/extra/wrong characters, no
↪  spelling/grammatical errors in Chinese/English). Score 10: 100%
↪  accurate; Score 1: massive errors or unrecognizable characters.

2. Text-Background Harmony (1-10): Evaluate visual coordination: (1)
↪  text avoids blocking the background's main subject (key figures,
↪  core graphics); (2) text color/transparency ensures clear
↪  contrast with the background (no blurring). Score 10: no
↪  blocking, perfect contrast; Score 1: complete blocking or
↪  unreadable due to poor contrast.

3. Text Presentation Quality (1-10): Evaluate text's own properties:
↪  (1) structural rationality (clear title/body hierarchy,
↪  compliance with reading habits, balanced spacing); (2) physical
↪  readability (appropriate font selection, suitable size, neat
↪  alignment). Score 10: clear structure, highly readable; Score 1:
↪  chaotic structure and physically unreadable.

4. Meaning Expression Adaptability (1-10): Evaluate meaning
↪  transmission: (1) key information is highlighted (via
↪  weight/color/size); (2) layout matches text's emotional tone
↪  (e.g., serious text uses rigorous typography); (3) text position
↪  aligns with the background's semantic context (e.g., "ocean
↪  protection" text near ocean elements). Score 10: amplifies
↪  meaning, matches tone, aligns with background semantics; Score
↪  1: contradicts meaning/tone or conflicts with background
↪  semantics.
```

## D IMPLEMENTATION DETAILS

**Training Infrastructure:** All experiments were conducted on 16 NVIDIA H200 GPUs.

**Cold-Start SFT Stage:** We utilized approximately 8K multi-round iterative reflection trajectories for training, with the following distribution: 2,359 two-round samples, 1,266 three-round samples, 2,030 three-round samples, and 2,537 four-round samples. Note that the number of rounds equals the number of tool calls plus one, as the final round serves as the confirmation output stage. Two-round data represents cases where the initial generation is already satisfactory, and for these samples, the first-round responses were not masked during cold-start SFT training. Training hyperparameters were set as follows: batch size of 64, learning rate of 1e-5, trained for 2 epochs.

**Reinforcement Learning Stage:** During RL training, we set the maximum number of tool calls to 4. The weights for $R_{\text{ocr}}$ and $R_{\text{svg}}$(denoted as $\alpha$) were set to 0.25, while the weight for format reward $R_{\text{format}}$ was set to 0.1. We prepared up to 32K samples for training, with early stopping based on reward metrics during RL training. We employed a strict on-policy training strategy with the following configuration: batch size of 64, 8 rollouts per sample, sampling temperature of 1.0, KL divergence coefficient of 1e-3, and learning rate of 1e-6. To maintain visual feature stability, we froze the vision tower parameters and fine-tuned only the LLM components.

**Ablation Experiments:** For the 40K-sample SFT baseline, we used a batch size of 128. All other training configurations for SFT and RL remained consistent with the above settings.

## E MORE QUANTITATIVE RESULTS

### E.1 COMPARISON WITH EXISTING METHODS

Table 5: Graphic quality metrics and OCR metrics on the crello test set.

| Method | Model | Param Size | OCR | | | Graphic | | | RM Score ↑ |
|---|---|---|---|---|---|---|---|---|---|
| | | | Char-P ↑ | Char-R ↑ | Char-F ↑ | $R_{ali}$ ↓ | $R_{ove}$ ↓ | $R_{com}$ ↓ | |
| MLLM | GPT-4o | - | 0.9385 | 0.8834 | 0.9101 | **0.0017** | 0.0036 | 19.9216 | 0.4427 |
| | Claude3.7 | - | 0.8766 | 0.8459 | 0.8610 | 0.0058 | 0.0205 | 21.0798 | **0.5880** |
| | Doubao-Seed-1.6 | 230B | 0.8998 | 0.8531 | 0.8758 | 0.0051 | 0.0192 | 21.5918 | 0.4876 |
| | Qwen2.5-VL | 72B | 0.8403 | 0.7847 | 0.8115 | 0.0044 | 0.0619 | 24.6138 | 0.2359 |
| | Qwen2.5-VL | 7B | 0.8831 | 0.6701 | 0.7620 | 0.0028 | 0.0256 | 30.9031 | 0.2116 |
| Image Edit | GPT-4o | - | 0.8766 | 0.8781 | 0.8774 | - | - | - | 0.5849 |
| | Qwen-Image-Edit | 20B | 0.7965 | 0.8720 | 0.8326 | - | - | - | 0.3671 |
| | FLUX Kontext | 12B | 0.5734 | 0.5393 | 0.5558 | - | - | - | -0.0834 |
| Layout | OpenCOLE | 7B | 0.7778 | 0.5453 | 0.6411 | 1.1902 | 0.0429 | 31.5831 | 0.3345 |
| Ours | Visual Feedback-step1 | 7B | 0.9407 | 0.8221 | 0.8774 | 0.0046 | 0.0061 | 19.5917 | 0.4392 |
| | Visual Feedback-answer | 7B | **0.9468** | **0.9054** | **0.9256** | 0.0025 | **0.0022** | **14.8063** | 0.5548 |
| | Δ (vs step1) | 7B | +0.0061 | +0.0833 | +0.0482 | +0.0021 | +0.0039 | +4.7854 | +0.1156 |

Table 6: The GPT-4o metrics on the crello test set.

| Method | Model | Param Size | GPT-4o | | | | |
|---|---|---|---|---|---|---|---|
| | | | Text | Harmony | Quality | Meaning | Overall |
| MLLM | GPT-4o | - | 8.9612 | 7.9949 | 7.5237 | 7.9618 | 8.1104 |
| | Claude3.7 | - | **8.9645** | 8.3805 | 7.8999 | 8.4507 | 8.4239 |
| | Doubao-Seed-1.6 | 230B | 8.8927 | 8.2174 | 7.6658 | 8.2569 | 8.2582 |
| | Qwen2.5-VL | 72B | 7.7332 | 7.4732 | 6.5589 | 6.5405 | 7.0764 |
| | Qwen2.5-VL | 7B | 7.6289 | 6.2840 | 5.5726 | 5.5813 | 6.2667 |
| Image Edit | GPT-4o | - | 8.8438 | **9.0127** | **8.8051** | **9.0069** | **8.9171** |
| | Qwen-Image-Edit | 20B | 5.7509 | 7.2808 | 6.0635 | 5.9903 | 6.2714 |
| | FLUX Kontext | 12B | 3.2447 | 6.6992 | 4.5005 | 4.1499 | 4.6486 |
| Layout | OpenCOLE | 7B | 6.9777 | 7.3864 | 6.4068 | 6.5319 | 6.8257 |
| Ours | Visual Feedback-step1 | 7B | 8.2334 | 7.9055 | 7.2046 | 7.6543 | 7.7494 |
| | Visual Feedback-answer | 7B | 8.8260 | 8.5957 | 7.7267 | 8.2964 | 8.3562 |
| | Δ (vs step1) | 7B | +0.5926 | +0.6902 | +0.5221 | +0.6421 | +0.6068 |

### E.2 ABLATION STUDY

### E.3 REWARD MODEL EVALUATION

We trained the reward model on a preference dataset constructed from 200K layout samples. Following the procedure in Section 3.3.2, we generated four quality levels (Level-I, Level-II, Level-III,

Table 7: Graphic quality metrics and OCR metrics on the DESIGNERINTENTION test set.

| Method | Model | Param Size | OCR | | | Graphic | | | RM Score ↑ |
|---|---|---|---|---|---|---|---|---|---|
| | | | Char-P ↑ | Char-R ↑ | Char-F ↑ | $R_{ali}$ ↓ | $R_{ove}$ ↓ | $R_{com}$ ↓ | |
| MLLM | GPT-4o | - | 0.9683 | 0.9447 | 0.9563 | 0.0015 | 0.0140 | 18.7249 | 0.5072 |
| | Claude3.7 | - | 0.8680 | 0.8828 | 0.8753 | 0.0070 | 0.0601 | 15.0869 | **0.6465** |
| | Doubao-Seed-1.6 | 230B | 0.9102 | 0.8739 | 0.8917 | 0.0046 | 0.0222 | 15.7470 | 0.5103 |
| | Qwen2.5-VL | 72B | 0.5421 | 0.5530 | 0.5475 | 0.0069 | 0.1649 | 17.8489 | 0.3104 |
| | Qwen2.5-VL | 7B | 0.8660 | 0.5455 | 0.6694 | **0.0011** | 0.0689 | 19.1119 | 0.2018 |
| Image Edit | GPT-4o | - | 0.8839 | 0.8825 | 0.8832 | - | - | - | 0.6278 |
| | Qwen-Image-Edit | 20B | 0.8332 | 0.8713 | 0.8518 | - | - | - | 0.5995 |
| | FLUX Kontext | 12B | 0.5623 | 0.5217 | 0.5412 | - | - | - | 0.4426 |
| Layout | OpenCOLE | 7B | 0.7924 | 0.6781 | 0.7308 | 0.6848 | 0.0408 | 20.2853 | 0.4684 |
| Ours | Visual Feedback-step1 | 7B | 0.9700 | 0.9146 | 0.9415 | 0.0033 | 0.0023 | 14.1230 | 0.5285 |
| | Visual Feedback-answer | 7B | **0.9781** | **0.9547** | **0.9663** | 0.0024 | **0.0008** | **12.1167** | 0.5688 |
| | Δ (vs step1) | 7B | +0.0081 | +0.0401 | +0.0248 | +0.0009 | +0.0015 | +2.0063 | +0.0403 |

Table 8: The GPT-4o metrics on the DESIGNERINTENTION test set.

| Method | Model | Param Size | GPT-4o | | | | |
|---|---|---|---|---|---|---|---|
| | | | Text | Harmony | Quality | Meaning | Overall |
| MLLM | GPT-4o | - | 9.5180 | 8.2725 | 8.0200 | 8.4820 | 8.5731 |
| | Claude3.7 | - | **9.5815** | 8.5473 | 8.2052 | 8.8229 | 8.7892 |
| | Doubao-Seed-1.6 | 230B | 9.1440 | 8.0020 | 7.7400 | 8.2960 | 8.2955 |
| | Qwen2.5-VL | 72B | 4.1891 | 7.7565 | 6.8813 | 4.4970 | 5.8310 |
| | Qwen2.5-VL | 7B | 7.3688 | 6.1285 | 5.6576 | 5.4498 | 6.1512 |
| Image Edit | GPT-4o | - | 9.5231 | **8.9287** | **9.0126** | **9.2558** | **9.1800** |
| | Qwen-Image-Edit | 20B | 7.1626 | 7.7435 | 7.2625 | 7.1403 | 7.3272 |
| | FLUX Kontext | 12B | 4.3908 | 7.2806 | 5.4208 | 5.1503 | 5.5606 |
| Layout | OpenCOLE | 7B | 8.4738 | 7.3488 | 6.9516 | 7.4435 | 7.5544 |
| Ours | Visual Feedback-step1 | 7B | 9.4020 | 8.1924 | 7.8640 | 8.3260 | 8.4461 |
| | Visual Feedback-answer | 7B | 9.5569 | 8.4511 | 7.9301 | 8.5130 | 8.6128 |
| | Δ (vs step1) | 7B | +0.1549 | +0.2587 | +0.0661 | +0.1870 | +0.1667 |

Table 9: Multiple ablation experiments on Graphic quality metrics and OCR metrics on crello test set.

| Model | OCR | | | Graphic | | | RM Score ↑ |
|---|---|---|---|---|---|---|---|
| | Char-P ↑ | Char-R ↑ | Char-F ↑ | $R_{ali}$ ↓ | $R_{ove}$ ↓ | $R_{com}$ ↓ | |
| Visual Feedback-step1 | 0.9407 | 0.8221 | 0.8774 | 0.0046 | 0.0061 | 19.5917 | 0.4392 |
| Visual Feedback-answer | **0.9468** | **0.9054** | **0.9256** | 0.0025 | 0.0022 | 14.8063 | **0.5548** |
| cold-start-step1 | 0.9140 | 0.6962 | 0.7904 | 0.0118 | 0.0410 | 24.3143 | 0.2721 |
| cold-start-answer | 0.9121 | 0.7011 | 0.7928 | 0.0121 | 0.0402 | 24.0888 | 0.2774 |
| cold-start-only-step1 | 0.9321 | 0.8387 | 0.8829 | 0.0010 | 0.0094 | 24.4845 | 0.4007 |
| Qwen2.5-VL-7B | 0.8831 | 0.6701 | 0.7620 | 0.0028 | 0.0256 | 30.9031 | 0.2116 |
| direct RL | 0.9230 | 0.8394 | 0.8792 | **0.0004** | **0.0012** | 30.3397 | 0.3482 |
| direct sft8k+rl | 0.9400 | 0.8803 | 0.9092 | 0.0009 | 0.0022 | 19.3983 | 0.4596 |
| basemodel-direct-40k | 0.9348 | 0.8604 | 0.8960 | 0.0032 | 0.0192 | 18.3754 | 0.4680 |

Table 10: Multiple ablation experiments on Graphic quality metrics and OCR metrics on DE-SIGNERINTENTION set.

| Model | OCR | | | Graphic | | | RM Score ↑ |
|---|---|---|---|---|---|---|---|
| | Char-P ↑ | Char-R ↑ | Char-F ↑ | $R_{ali}$ ↓ | $R_{ove}$ ↓ | $R_{com}$ ↓ | |
| Visual Feedback-step1 | 0.9700 | 0.9146 | 0.9415 | 0.0033 | 0.0023 | 14.1230 | 0.5285 |
| Visual Feedback-answer | **0.9781** | **0.9547** | **0.9663** | 0.0024 | **0.0008** | **12.1167** | **0.5688** |
| cold-start-step1 | 0.9546 | 0.8266 | 0.8860 | 0.0130 | 0.0264 | 17.5229 | 0.3784 |
| cold-start-answer | 0.9548 | 0.8301 | 0.8881 | 0.0127 | 0.0257 | 16.9280 | 0.3850 |
| cold-start-only-step1 | 0.9666 | 0.9115 | 0.9382 | 0.0006 | 0.0033 | 16.7466 | 0.4768 |
| Qwen2.5-VL-7B | 0.8660 | 0.5455 | 0.6694 | 0.0011 | 0.0689 | 19.1119 | 0.2018 |
| direct RL | 0.9618 | 0.9250 | 0.9430 | **0.0005** | **0.0001** | 18.0272 | 0.4692 |
| direct sft8k+rl | 0.9752 | 0.9433 | 0.9590 | **0.0005** | 0.0010 | 14.6338 | 0.5167 |
| basemodel-direct-40k | 0.9544 | 0.9327 | 0.9434 | 0.0017 | 0.0082 | 13.1702 | 0.5398 |

Table 11: The performance of the four quality-level datasets in terms of OCR and Graphic metrics, as well as the scores from our trained reward model: The OCR scores and Graphic metrics indicate that our four-level data have a good hierarchical progressive relationship; the RM score shows that our trained reward model can well distinguish between quality levels.

| leval | OCR | | | Graphic | | | RM Score ↑ |
|---|---|---|---|---|---|---|---|
| | Char-P ↑ | Char-R ↑ | Char-F ↑ | $R_{ali}$ ↓ | $R_{ove}$ ↓ | $R_{com}$ ↓ | |
| Level-I | 0.9752 | 0.9211 | 0.9474 | 0.0089 | 0.0038 | 6.6795 | 1.0594 |
| Level-II | 0.9644 | 0.8522 | 0.9049 | 0.0112 | 0.0134 | 12.2626 | 0.6343 |
| Level-III | 0.8676 | 0.4197 | 0.5657 | 0.0354 | 0.0444 | 21.2628 | -0.2345 |
| Level-IV | 0.8478 | 0.3880 | 0.5324 | 0.0536 | 0.0375 | 17.5270 | -1.3457 |

Level-IV) for each query, yielding 1.2M preference pairs. We randomly selected 25K pairs as the test set, using the remainder for training. During training, we use a batch size of 512 and train for 2100 steps.

To stabilize the reinforcement learning process, the raw output of our trained reward model, $r_\theta$, is normalized before being used as the final layout reward, $R_{\text{layout}}$. Following the practice in Xu et al. (2023), we first compute the distribution of $r_\theta$ scores across the test set. The scores are then standardized using the mean and standard deviation of this distribution. This procedure ensures that the reward signal maintains a consistent scale throughout training, which is crucial for effective policy optimization. The final reward is calculated as:

$$R_{\text{layout}} = \frac{r_\theta - \text{mean}_{\text{test}}(r_\theta)}{\text{std}_{\text{test}}(r_\theta)} \tag{11}$$

**Results:** The trained reward model achieves a high pairwise prediction accuracy of 97.4% on the test set, demonstrating its strong ability to discern finer-grained layout preferences.

To further validate our methodology, we conducted two key analyses presented in Table 11. First, we verify the integrity of our four-level data hierarchy using objective metrics. As shown, the external Graphic and OCR metrics exhibit a clear monotonic degradation from Level-I to Level-IV. This result confirms that our data construction process successfully creates a well-defined and reliable quality gradient.

Second, we evaluated whether our trained reward model internalizes this quality structure. The final column of the table reports the average Reward Model (RM) Score, i.e., $R_{\text{layout}}$ for test samples at each level. The RM scores align perfectly with the established hierarchy, decreasing consistently from a high for Level-I to a low for Level-IV. This strong discriminative performance across distinct quality levels confirms that our model has learned a nuanced understanding of layout quality, enabling it to provide a reliable and effective supervision signal for reinforcement learning.

## F   MORE QUALITATIVE RESULTS

The comparison of data generated on all baselines is shown in Figure 4.

## G   PROMPT FOR DATA CONSTRUCTION

```
Initial Reasoning Process Prompt

Role setting:
You are an experienced Layout and SVG engineer.

Task:
Here is a result of using SVG code to typeset specific text on an
↪  input background image. I will provide you with the designed SVG
↪  code and the rendered image of this code, which has a very
↪  beautiful layout effect.
Now, assuming you are the designer of typesetting this SVG, what is
↪  your thought process when typesetting this SVG?
Could you please use the voice of a designer to briefly describe
↪  your thought process when designing this SVG based on the SVG
↪  code and rendering results? How did you design this SVG?
```

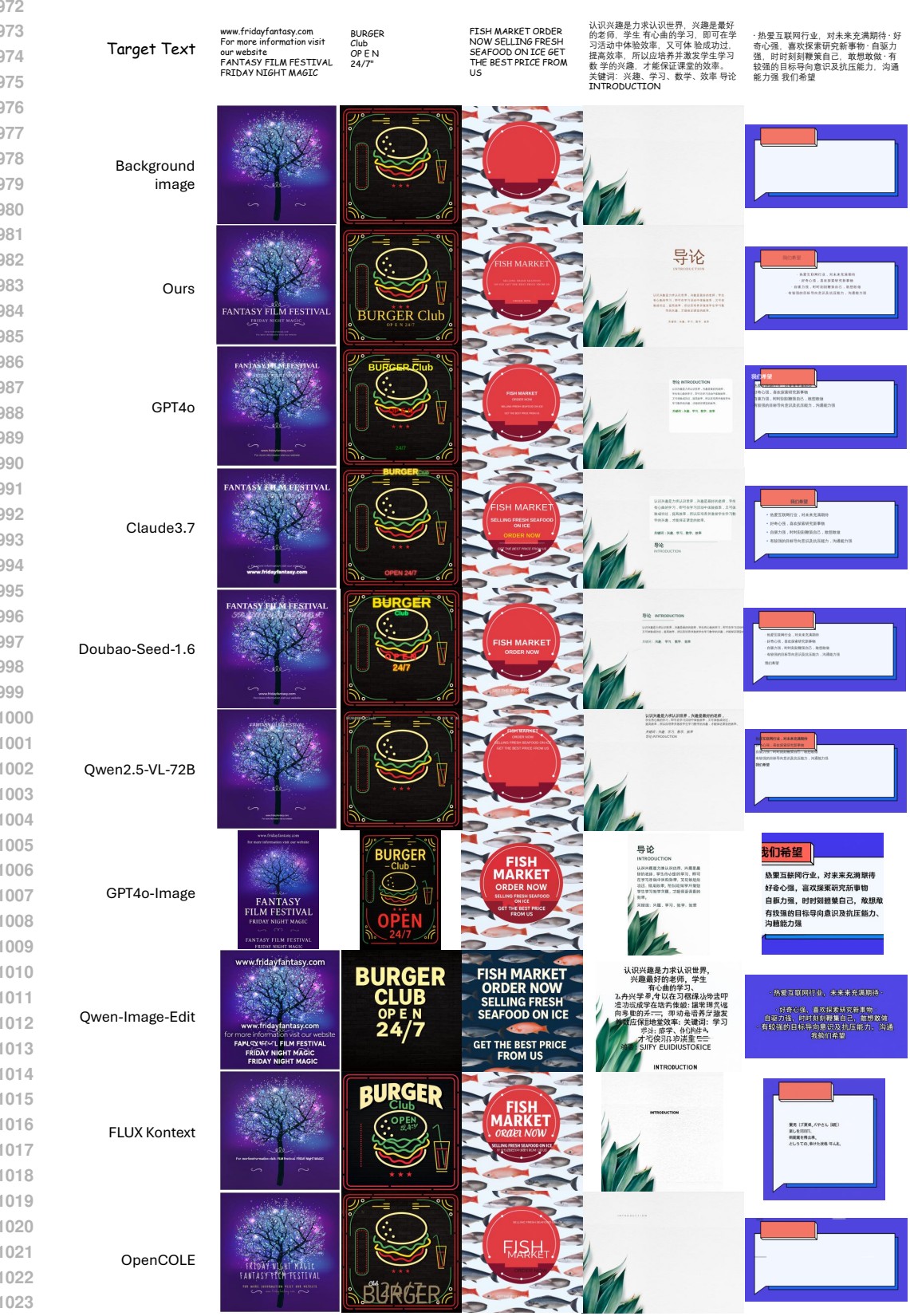

Figure 4: In comparison with all existing methods.

```
1026
1027    Ensure that your design ideas are consistent and closely related to
1028  ↪   the design results of this SVG. Do not fabricate content that is
1029  ↪   not included in the SVG, as the SVG only typesets the given text
1030  ↪   based on the given background image. Therefore, the typesetting
1031  ↪   ideas should focus on the implementation of the text, including
1032  ↪   text position, font, size, beautification effect, etc. Don't pay
1033  ↪   attention to the text already in the background image. Do not
1034  ↪   pay attention to unnecessary other graphic icons and other
1035  ↪   elements, and do not analyze how good the rendered image is from
1036  ↪   the perspective of typesetting, because your thinking is based
        ↪   on not seeing the rendered image.

1037    background-image.png: <image>
1038    image size : {image_width} * {image_height}
1039    texts: {target_text}
1040
1041    svg code:
1042    ```svg
1043    {svg_code}
1044    ```
1045    The image result rendered by this svg: <image>

1046    Output requirements:
1047    Please directly output the thinking process of the person in the
1048  ↪   designer's tone, without any other descriptive content. Answer
        ↪   in English.
1049
1050
1051
1052    Multi-Round Reflection Data Synthesis Prompt
1053
1054    USER:
1055
1056    You are an experienced typesetter and SVG engineer, skilled at
1057  ↪   elegantly typesetting specified text on user provided background
        ↪   images.
1058    You know how to apply unique aesthetic principles to design
1059  ↪   professional and attractive layout, using SVG code to create
1060  ↪   beautiful layouts. Please design the final layout plan based on
1061  ↪   the background image and text content provided by the user.
1062    In SVG code, use the image tag to reference the background image:
1063  ↪   href=\"background-image.png\", while other elements only need to
1064  ↪   be designed with content related to the text.
1065    Please design an SVG code layout scheme based on the background
        ↪   image and text content provided by the following users.
        You first think about it, and then output the final SVG code. The
1066  ↪   format is<think>...</think>\n<answer>...</answer>
1067
        background-image.png: <image>
1068    image size: {image_width}*{image_height}
1069    Please arrange the following text on the background image:
1070    {target_text}
1071
        ASSISTANT:
1072    {Reasoning Process and SVG generated by Qwen2.5-VL-7B}
1073
1074    USER:
1075    Now, I'll show you the effect of this version of SVG layout, and you
1076  ↪   need to improve this SVG layout effect.
        I'll also give you a standard SVG layout result, and you need to
1077  ↪   improve your SVG layout according to this standard SVG layout
1078  ↪   result.
1079
```

```
1080
1081    - In your output, you need to speak in the tone of a designer,
1082   ↪  stating that you've reviewed the SVG result of your initial
1083   ↪  layout, then reflected on it and made corrections. Note that
1084   ↪  you've designed an initial version of the SVG, and now I've
1085   ↪  provided you with the rendered image. Your output should focus
1086   ↪  on examining the image, ensuring it's a reflection and
1087   ↪  correction of your initial SVG layout result. The direction of
1088   ↪  correction is the correct effect I gave you, but don't expose in
1089   ↪  the output that you're improving based on the standard effect.
1090   ↪  Pretend you've thought it out on your own.

1091    - The output should include your thinking process for SVG layout,
1092   ↪  how to improve your SVG layout result step by step. You need to
1093   ↪  point out which parts of your initial layout were good and which
1094   ↪  were bad and needed modification. For each modification point,
1095   ↪  be specific about how to modify the SVG code. Don't just
1096   ↪  qualitatively say which aspects you'll modify. Pay attention to
1097   ↪  the tone, which should be like that of a designer, and the
1098   ↪  content of the output should conform to the designer's way of
1099   ↪  thinking.

1100    - During the modification process, key considerations should be text
1101   ↪  position, whether there is any text missing, text overlapping,
1102   ↪  text being blocked, and whether the text exceeds the background
1103   ↪  image range, etc. These considerations need to be included in
1104   ↪  the output.

1105    - Your output modification process may involve multiple steps. If
1106   ↪  your initial layout is not very different from the standard one,
1107   ↪  you can make only one modification; if there is a large gap,
1108   ↪  multiple steps of modification are required. You need to
1109   ↪  simulate the designer's thinking process and gradually improve
1110   ↪  the SVG layout. Each time you modify, choose the part with the
1111   ↪  worst effect to improve. Explain the specific SVG code
1112   ↪  improvements in the thinking process. After modifying one
1113   ↪  version, only make changes to the SVG part that needs to be
1114   ↪  modified in this step, and don't change the other parts for now.
1115   ↪  Output the complete SVG code; then proceed to the next
1116   ↪  modification until you think the SVG layout effect is very good.
1117   ↪  Don't make too many modifications. Ensure that each modification
1118   ↪  is better than the previous one, with a maximum of 3
1119   ↪  modifications. The SVG code after the last modification needs to
1120   ↪  be output, and its effect should be the same as that of the
1121   ↪  standard code I gave you.

1122    - You need to answer one modification each time, and then I'll show
1123   ↪  you the rendered effect of the SVG you modified, and you'll make
1124   ↪  the next modification.

1125    - Based on the rendered image effect I give you after each of your
1126   ↪  modifications, decide whether the next modification is needed.
1127   ↪  Each modification should have a significant improvement, not
1128   ↪  just a minor one. For example, when the order of different text
1129   ↪  tags doesn't affect the SVG rendering effect, there's no need
1130   ↪  for additional modification. Since I require you to make as few
1131   ↪  steps of modification as possible, each modification should have
1132   ↪  a significant improvement.
1133
```

```
Your output is the thinking process of a designer improving the SVG
↪   layout after reviewing the first version they designed. I've
↪   given you the standard SVG code, and you should modify the SVG
↪   code in this direction. However, note that your output is based
↪   on not having seen this standard SVG, as if the designer is
↪   reflecting after designing the initial draft and modifying it to
↪   the final standard SVG version through multiple steps.

After the last modification, you need to output the final
↪   inspection, indicating that after checking the image, you think
↪   the current SVG layout effect is very good and can be replied to
↪   the user.

Your initial SVG layout effect is shown in the figure below:
<image>
This is standard and beautiful SVG code. The code and its rendered
↪   effect diagram are as follows.
```svg
{svg_code}
```
<image>

Output requirements:
Please directly output the thinking process in the tone of a
↪   designer, without any other descriptive content. Be careful not
↪   to reveal that you have seen the standard SVG effect. Transform
↪   it into your own thinking. The output should conform to the
↪   designer's thinking process, that is, how you think about
↪   improving the layout by yourself, not by comparing with the
↪   standard effect. Do not output the word "standard".

If improvement is needed, the first sentence in each step of the
↪   thinking process should be: "I will check the SVG rendering
↪   effect of my version...", and the last sentence should be:
↪   "Next, I will improve my SVG code."

These two beginning and ending sentences are necessary and cannot be
↪   omitted, but you can modify the language to maintain the same
↪   meaning and make the output diverse.

Your output needs to specifically point out which effects in your
↪   first version are good and do not need improvement, which
↪   effects are poor and need improvement, and how to specifically
↪   modify the SVG code. If you think the SVG layout effect of your
↪   first version is very close to or even better than the standard
↪   SVG rendering effect I provided, you can describe your
↪   satisfaction with this SVG layout and that you think it has
↪   achieved a very good effect and does not need further
↪   improvement.

Answer in English.

Output requirements:
- You need to output in the form of multi – round conversations.
↪   According to the number of modifications you decide, the output
↪   format for each modification is as follows:

# Step {current modification number} of modification:

## Thinking process: Here, think about how to make the modification.

## SVG code: Modify the complete SVG code.
```

```
- After the final modification, the rendering result of your SVG
↪   code should be exactly the same as that of the reference SVG
↪   code I provided.

- After the last modification is output, I will provide you with the
↪   rendered image again. Then you need to output a final
↪   reflection, indicating that you will check the SVG rendering
↪   effect of this version and think that the current SVG layout
↪   effect is very good and does not need to be improved further,
↪   and it can be output to the user. The output format of the final
↪   reflection is:

# Final rethink: ...

USER:
Your current SVG layout effect is shown in the figure below:
<image>

ASSISTANT:
...
```

