# OpenReview forum: "Visual Feedback for Self-Improving Text Layout with MLLM via Reinforcement Learning"
_ICLR.cc/2026/Conference — ICLR 2026 Conference Withdrawn Submission_

### Official Review · Reviewer_jKJr · 2025-10-30

**Soundness:** 3
**Presentation:** 3
**Contribution:** 3
**Rating:** 6
**Confidence:** 3

**Summary:**

The paper presents a visual-feedback, iterative text-layout framework for multimodal large language models (MLLMs). The framework operates by generating SVG code, rendering it into an image, visually inspecting the result, and iteratively refining the output. Training involves supervised fine-tuning (SFT) on teacher-distilled multi-round traces, followed by GRPO optimization with a hybrid reward function. The overarching objective is to produce layouts that more reliably satisfy spatial constraints and ensure text readability (as measured by OCR accuracy), outperforming existing code-only generation pipelines.

**Strengths:**

1. The proposed closed-loop generation–rendering–reflection–refinement cycle is both creative and effective, providing a compelling solution to the visual perception gap in text-to-layout generation.

2. The paper offers an insightful observation that a final-outcome reward can drive self-improvement more effectively than complex process-level reward designs, supported by clear ablation studies and informative learning curves.

3. The introduction of a hierarchical four-level preference dataset represents a practical and scalable strategy for training a layout reward model capable of fine-grained discernment.

**Weaknesses:**

1. While the paper reports the performance difference between step 1 and the final output (“answer”), it does not provide the distribution of rounds-to-convergence or analyze whether excessive iterations may lead to quality degradation. Such analysis would offer valuable insight into the stability and efficiency of the iterative refinement process.

2. The current description is insufficient to evaluate dataset diversity and reproducibility. Key details are missing, including data sources, cleaning and filtering criteria, distributions of language and font types, text-length buckets, and background complexity. Adding a dataset card and detailed preprocessing documentation—either in the main text or appendix—would significantly improve transparency and replicability.

3. The paper also lacks a discussion of efficiency and inference cost. Reporting runtime statistics or computational overhead per iteration would help readers assess the practicality of the proposed approach.

**Questions:**

Baseline coverage for “layout” methods feels thin (only OpenCOLE). Could you include stronger, more recent baselines such as PosterO?

[1] H.-Y. Hsu and Y. Peng. PosterO: Structuring Layout Trees to Enable Language Models in Generalized Content-Aware Layout Generation. In Proc. CVPR, 2025. arXiv:2505.07843.

---

### Official Review · Reviewer_xuuY · 2025-10-31

**Soundness:** 3
**Presentation:** 2
**Contribution:** 2
**Rating:** 6
**Confidence:** 4

**Summary:**

This paper introduces a visual feedback-driven, self-improving framework for text-layout generation using multimodal LLMs (MLLMs). Instead of the typical “text-to-code-only” paradigm (where layout code is produced and rendered blindly), the authors close the loop: the model generates SVG code, renders it, visually inspects the result, and refines it iteratively. A two-stage pipeline is proposed, SFT from a synthetic multi-round dataset, followed by RL using a visually grounded reward model that combines OCR accuracy, layout aesthetics, and format correctness.
Experiments on several layout benchmarks show substantial improvements over state-of-the-art MLLMs (e.g., GPT-4o, Claude-3.7) and open-source layout models (e.g., OpenCOLE). The paper also reports an insightful finding that simple outcome-based rewards outperform more complex process-oriented rewards in stimulating self-improvement behavior.

**Strengths:**

1. The generation–render–reflection–refinement loop is conceptually simple yet powerful, integrating visual inspection into code generation in a fully automated way.

2. The paper ablates SFT vs. RL stages, dataset scales, and reward formulations, showing convincing incremental improvements.

**Weaknesses:**

1. The reflection data are entirely generated by Doubao-Seed-1.6. This may introduce stylistic bias or restrict the diversity of layout reasoning.

2. While Eq. (6) describes pairwise preference learning, there is no ablation on how sensitive RL performance is to reward scaling or weighting (α, γ).

3. Since the framework performs multiple render–feedback iterations, inference cost may be high. The paper does not quantify average iteration count, GPU time, or convergence behavior—key factors for practical adoption.

4. The paper should provide some failure cases, where visual feedback still fails (e.g., cluttered text, small fonts) and why.

**Questions:**

1. How many reflection rounds are typically needed before the model declares satisfaction? Does this vary with input complexity?

2. Could the framework handle multilingual or non-Latin text layouts, where OCR accuracy drops sharply?

3. What happens if the rendered layout contains transparent text or artistic fonts—does the OCR reward still work reliably?

4. Could this self-improving loop be extended to image composition tasks, e.g., arranging visual objects, rather than text boxes?

5. Are the visual feedback iterations differentiable through the renderer, or purely discrete (black-box RL)?

6. How large is the reward-model training set (number of triplets/pairs)? Can it generalize to unseen background styles?

---

### Official Review · Reviewer_4Dsj · 2025-11-05

**Soundness:** 2
**Presentation:** 2
**Contribution:** 2
**Rating:** 4
**Confidence:** 4

**Summary:**

This work highlights the importance of applying self-refinement through visual feedback in layout generation models. For this, authors first perform a cold-start supervised fine-tuning of Qwen2.5-VL-7B using multi-stage generation–reflection–refinement trajectories generated using Doubao-Seed-1.6. They then further train the model with GRPO reinforcement learning, using as a reward that combines: (1) the score from a Qwen2.5-VL-3B model trained to evaluate layout quality, (2) text accuracy measured through OCR, and (3) accuracy of the text content in the produced SVG compared against the target text.

Results on their test sets, their model outperforms large general-purpose multimodal LLMs and a layout-generation baseline. Through an ablation study they show that visual feedback is the main performance driver, while RL provides an additional (but smaller) improvement.

**Strengths:**

The method outperforms both open-weight and commercial multimodal models, many of which have significantly more parameters, and it clearly surpasses the domain-specific layout generation baseline.

They highlight an interesting result showing that simple outcome-based rewards (final layout quality, OCR accuracy, and SVG accuracy) are enough to maintain iterative self-improvement and outperform more complex process-based rewards.

The ablation study shows that the visual feedback loop itself, not just RL, is the main driver of model performance.

**Weaknesses:**

The method applies an existing mechanism (self-refinement / visual feedback + RL) to a narrow and application-specific task. While the findings are significant for this area, the general contribution may be borderline for ICLR. Transferring the findings of this work to other tasks would strengthen the paper.

The comparison to the only other layout-generation system (OpenCOLE) is potentially unfair. OpenCOLE is evaluated zero-shot on a test set sampled from the authors' own dataset, while the proposed model is explicitly trained on it. To support the claim Visual Feedback is a critical factor in layout generation, OpenCOLE should ideally be fine-tuned on the same training set. While the proposed model remains over-performant, the performance gap agaisnt OpenCOLE is significantly reduced when tested in the crello and DESIGNERINTENTION test sets.

**Questions:**

I would suggest including the GRPO objective in a single line (following Shao et al., 2024), moving the full equation to the appendix, or directly refer to Shao et al., 2024. The same suggestion applies to Equation (6) (log-likelihood loss).

The four layout quality levels might be too easily separable, given the high reward-model accuracy. How well does the reward model score correlate with human judgments?

In Table 3 (ablation study), the model variants should be clearly referenced to their definitions in the paragraph.

Figure 3 should improve readability (the text is too small).

---

### Official Review · Reviewer_VitP · 2025-11-08

**Soundness:** 2
**Presentation:** 2
**Contribution:** 2
**Rating:** 4
**Confidence:** 4

**Summary:**

The paper studies text layout generation by incorporating visual feedback into the generation process, which repeats the loop of <generation, rendering, reflection, refinement>. The training employs a standard SFT+RL framework. The work also includes the construction of required data, i.e., suboptimal layout, iterative reflection, modification, etc. A reward model is developed to provide the layout quality score in GRPO training. In the experiments, the authors compare their method against MLLMs, image editing models, and layout generation models to show its superior performance.

**Strengths:**

The motivation of the paper is natural and intuitive. By rendering the step 1 layout and examining the image quality, the model can make continuous progress towards a good text layout. The experimental results demonstrate the effectiveness of visual feedback in text layout generation. The finding that simple outcome rewards can be more effective than complex process rewards challenges the common sense in agentic RL.

**Weaknesses:**

- The implementation details of the MLLM baselines are not included. What is the prompt used for these baselines? Do they also have a complete visual feedback process to rectify their initial outputs?
- The paper should include the discussion and comparison with more specialized layout generation models, e.g., PosterLlama [1], PosterLLaVa [2], LaDeCo [3], FlexDM [4], etc.
- Efficiency analysis is not included in the experiments. The approach includes multiple steps, how does its efficiency lag behind other methods? Also, there is a lack of analysis on the refinement rounds. On average, how many rounds are needed to achieve stable results? Are there any ablation studies on round number?
- No human evaluation to verify that the improvements from visual feedback are perceptually meaningful.
- Is the OpenCOLE baseline fine-tuned on the same training set? If not, the comparison may be unfair.


[1] Seol, Jaejung, Seojun Kim, and Jaejun Yoo. "Posterllama: Bridging design ability of langauge model to contents-aware layout generation." arXiv preprint arXiv:2404.00995 (2024).

[2] Yang, Tao, et al. "Posterllava: Constructing a unified multi-modal layout generator with llm." arXiv preprint arXiv:2406.02884 (2024).

[3] Lin, Jiawei, et al. "From Elements to Design: A Layered Approach for Automatic Graphic Design Composition." Proceedings of the Computer Vision and Pattern Recognition Conference. 2025.

[4] Inoue, Naoto, et al. "Towards flexible multi-modal document models." Proceedings of the IEEE/CVF Conference on Computer Vision and Pattern Recognition. 2023.

**Questions:**

1. What is the layout dataset used for model training? Is it a publicly available dataset, e.g., Crello, or a proprietary dataset?
2. RL training includes 3 different rewards. How do these rewards improve the qualitative and quantitative results, respectively? Are there any ablation studies on reward design?
3. Could the authors visualize the intermediate results in the generation process as well as the step-wise feedback?

---

### Note · Authors · 2025-11-13

**Comment:**

We would like to express our sincere gratitude to all reviewers for their insightful comments and constructive suggestions. We highly value each piece of feedback and intend to incorporate them to refine the paper in our future work. At this time, we have decided to withdraw the manuscript.

We wish you continued success in your future research endeavors.

**Withdrawal Confirmation:**

I have read and agree with the venue's withdrawal policy on behalf of myself and my co-authors.